# Research on the emissions from industrial products exported from Guangdong Province —an input-output model analysis

**Yu-Xia Du**[1,2,3]**, Ming-Jie Li**[1,3,4]***, Jun-Jie Huang**[1]

**1** Guangzhou Huashang College, Guangzhou, China, **2** Guangdong University of Foreign Study, Guangzhou, China, **3** Institute for Economic and Social Research, Guangzhou Huashang College, Guangzhou, China, **4** Central South University of Forestry and Technology, Changsha, China

* changfenglmj@163.com

**Data Availability Statement:** All relevant data are within the paper and its Supporting Information files.

## Abstract

This study uses an input-output model to analyze the wastewater, waste gas, and solid waste emissions in Guangdong's industrial exports from 2004 to 2015; the Logarithmic Mean Divisia Index (LMDI) is used to analyze the factors influencing such pollution. The results reveal that embodied emissions of waste gas and solid waste in Guangdong's export trade are increasing, while the increase in wastewater emissions is not apparent. The Logarithmic Mean Divisia Index (LMDI) is used to analyze the influencing factors of pollution, specifically, the structural, scale, and technical effects. We discovered that emissions of the top five industries account for about 80% of total emissions and the wastewater emissions' technical effect has more impact; however, it is difficult for this technical effect in terms of embodied waste gas and solid waste to offset the scale and structural effects' impacts. Moreover, the trends and factors influencing various industries' pollution emissions differ. This study proposes that when the government carries out environmental pollution control measures, they should consider the embodied pollution caused by products from foreign trade and focus on treating industries with severe pollution. Simultaneously, the pollution controlling measures of different industries should also vary.

## 1. Introduction

High-quality economic development requires economic benefits as well as environmental protection. China's President Xi Jinping has emphasized the construction of an ecological civilization in his many speeches. On April 30, 2021, at the 29th Collective Study Conference of the Political Bureau of the 19th CPC Central Committee, he again emphasized that ecological environmental protection and economic development are dialectically unified and complementary. While China's foreign trade brings substantial economic benefits, it also consumes significant energy and creates high pollution emissions. Moreover, producing industrial trade products directly consumes resources and generates polluting emissions. Such production also involves intermediate products, which indirectly consume resources and create pollution during the production process.

**Funding:** This research was support by following fund which we thank for. And the authors would take all responsibility for this paper. 1.Huashang College Guangdong University of Finance & Economics: Research on the Ecological Footprint of Foreign Trade in Guangdong Province Based on Emergy Improvement. (Project Number: 2018HSXS08). The recipient is DU YU-XIA, who is the first author of this manuscript. The founder's contributor roles are conceptualization, data curation, formal analysis, methodology and project administration. 2. Guangzhou Huashang College: International Business Construction and Development (Project Number: HS2019CXQX17). The funders had no role in study design, data collection and analysis, decision to publish, or preparation of the manuscript. 3. Guangzhou Huashang College: Research team funding projects (Project Number:2021HSKT04). The funders had no role in study design, data collection and analysis, decision to publish, or preparation of the manuscript.

**Competing interests:** The authors have declared that no competing interests exist.

Guangdong Province is at the forefront of the country's reform and opening-up movement. Its total foreign trade has always accounted for approximately one-fourth of the country's foreign trade. Its primary imported and exported products are manufactured products, which account for more than 90% of its export products. Although industrial product trades bring economic benefits, the production process generates significant pollution emissions. If the embodied emissions are ignored, the economic benefits brought by trade will be overestimated, which will make it difficult to solve the actual environmental pollution problem brought by trade. This study analyzes the emissions produced in the trade process by using an input-output model to measure the complete pollution in Guangdong's foreign trade from 2004 to 2015; it also examines the influencing factors of these emissions to reveal the hidden pollution in foreign trade activities. We aim to provide helpful suggestions to further pollution reduction for Guangdong Province and other regions to decrease pollution and promote the green transformation of economic and social development while achieving high-quality development.

The innovation of this study is mainly in three aspects, which are as follows: (1) This study examines the embodied pollution from industrial products exported from Guangdong Province, and the research objects are wastewater, waste gas, and solid waste. (2) This study conducts the structural decomposition of the emissions from industrial products exported from Guangdong Province to find the impact mechanism involved. (3) The differences in embodied pollution in different industries are analyzed, so we can provide support to the government to implement differentiated policies for pollution control in various industries.

## 2. Research area and literature review

### 2.1 Overview of the research area

Guangdong Province is a prosperous economic province dominated by manufacturing and tertiary industries. It accounts for approximately 10% of China's total GDP, and its foreign trade for 25% of the country's foreign trade. Its foreign trade products involve primarily industrial products. Specifically, its proportion of exported industrial products is greater than 97%, and that of imported industrial products is approximately 90%. As a surplus state and from an economic perspective, the export of industrial products significantly contributed to the economic development of Guangdong Province. However, the production of exported industrial products also significantly impacted the domestic ecological environment. Fig 1 illustrates Guangdong Province's foreign trade from 2004 to 2015.

### 2.2 Literature review

Environmental problems generally occur when unreasonable development and resource utilization lead to ecological damage and environmental pollution. Externalities and market failures are believed to be economic causes of environmental problems. Leontief et al. [1] used the input-output approach to analyze air pollution and the economic structure. Grossman et al. [2] posited that trade primarily impacts the environment through scale, structural, and technological effects. Their "race to the bottom line" and "polluter's paradise" hypotheses describe trade's impact on the environment of various countries from an environmental regulatory perspective. Wyckoff et al. [3] estimated the amount of carbon embodied in the imports of manufactured goods to six of the largest OECD countries. Copeland et al. [4] studied the free trade impact on emissions reductions in northern and southern countries. Jaff et al. [5] examined environmental regulations' impacts on competitiveness among American industries. Peters [6] accurately determined environmental impacts from pollution embodied in the trade of Norway. Nathaniel et al. [7] studied the link between economic growth, energy use,

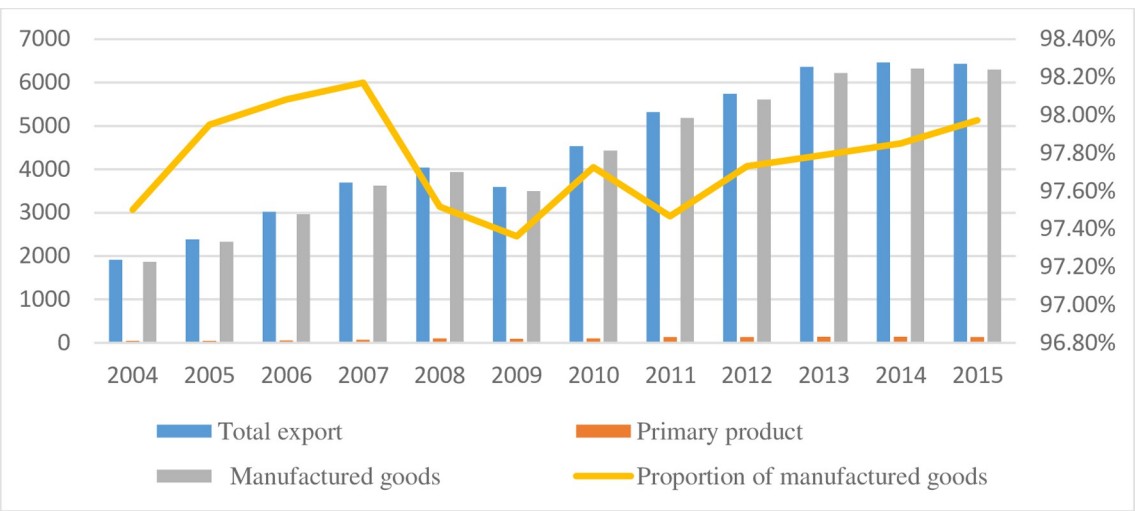

**Fig 1. The structure of exports in Guangdong Province (unit: USD 100 million).**

international trade, and the ecological footprint. Kim et al. [8] used MRIO (Multilateral regional input-output) to study the carbon dioxide emissions from international trade production and consumption in South Korea from 2000 to 2014. Dardati et al. [9] found that the intensity of exporters' emissions relates to the output measurement method, and exporters' outsourcing activities also impact the emissions situation. Tsagkari [10] evaluated carbon dioxide emissions embodied in international trade in Poland by the input-output approach.

Jiang et al. [11] analyzed the structural decomposition analysis (SDA) of global carbon emissions using the SDA method. Vinuya et al. [12] decomposed the growth in US $CO_2$ emissions state-wise by using the Logarithmic Mean Divisia Index (LMDI) method. Tunç [13] analyzed the decomposition of $CO_2$ emissions from energy use in Turkish. Alajmi [14] exposed the factors that affect GHG emissions in nine sectors of Saudi Arabia via the LMDI method from 1990 to 2016. So we can know the Logarithmic Mean Divisia Index(LMDI) is widely used to analyze the decomposition of emissions.

Fu et al. [15] theoretically and empirically analyzed the internalization of environmental costs and compared the RCA index of pollution-intensive industries in developed and developing countries to gain further insights. Li et al. [16] used the MRIO model to analyze the 2012 transfer of air pollution emissions given China's trade with other countries. Huang et al. [17] analyzed the pollution transfer problem inherent in the relationship between China and major partner countries. Shang [18] analyzed the environmental impacts of China's foreign trade for given four effects—scale, openness, structure, and technology—and pollution from trade in particular. Ma et al. [19] also reviewed pollution in China's foreign trade focusing on industrial products. Zhang et al. [20] analyzed the pollution within China's industrial product trade and its influencing factors. Zhang [21] calculated the carbon embodied in China's trade from 1987 to 2007 based on non-competitive (import) input-output tables. Ma et al. [22] calculated the implied energy consumption and carbon emissions found in the relationships between China and Regional Comprehensive Economic Partnership countries' import and export trades in 2015; the authors then decomposed the influencing factors of implied energy consumption and related carbon emissions for that year. Yuting et al. [23], Chen et al. [24], and DuGu [25] used the input-output method to analyze implied trade pollution. These literatures examined foreign trade emissions to reveal that the input-output method can better analyze the total amount of pollution emitted by the final product in each link of production;

moreover, this method can better reflect the environmental impacts from the product's entire production process. Regional and industrial pollutions also received extensive concern.

Single regional Input-output (SRIO) \ Bilateral trade input-output (BTIO) \Multilateral regional input-output (MRIO) is widely used to describe the emissions of trade, since this study examined the problem of Guangdong Province, so we used SRIO to analyze the emission of industrial products. In the decomposition of pollution emission structure, some scholars use the SDA decomposition method; in the process of SDA decomposition, there will be problems with interaction terms, such as inconsistent measurement results, poor comparability of factor weights, and difficulty in decomposing interaction effects. Since LMDI can overcome the cross-term problem well, LMDI is more widely used. The current study considers these results using the input-output method to analyze the pollution of industrial product exports in Guangdong Province. We also calculated the pollution observed in the industrial product trade of Guangdong Province from 2004 to 2015. While Guangdong Province's export trade products provide economic benefits, questions remain regarding the quantity of pollutants discharged from producing these foreign-trade industrial products and their extent of environmental impacts. We also use the LMDI index to analyze the scale, structure, and technological effects and indicate the impact of Guangdong's foreign trade products on the environment. Ultimately, our results provide a reference for transforming and upgrading of Guangdong's foreign trade.

## 3. Model and data

### 3.1 The input-output model

A country's input-output model is written as:

$$AX + Y = X \tag{1}$$

$$X = (I - A)^{-1}Y \tag{2}$$

where $A$ is the direct consumption coefficient matrix, also known as the technology matrix; $a_{ij} = x_{ij} / x_j$ represents the direct consumption of product $i$ for the unit output of $j$; and $X$ and $Y$ indicate the total and final product column vectors, respectively.

$$\begin{pmatrix} a_{11} & \cdots & a_{1n} \\ \vdots & \ddots & \vdots \\ a_{n1} & \cdots & a_{nn} \end{pmatrix} \cdot \begin{pmatrix} x_1 \\ \vdots \\ x_n \end{pmatrix} + \begin{pmatrix} y_1 \\ \vdots \\ y_n \end{pmatrix} = \begin{pmatrix} x_1 \\ \vdots \\ x_n \end{pmatrix} \tag{3}$$

The complete consumption coefficient matrix is denoted by $B$:

$$B = (I - A)^{-1} - I \tag{4}$$

$$B = \begin{pmatrix} b_{11} & \cdots & b_{1n} \\ \vdots & \ddots & \vdots \\ b_{n1} & \cdots & b_{nn} \end{pmatrix} \tag{5}$$

### 3.2 Calculation of emissions coefficients in the industrial sector

This study constructs a pollution coefficient for the industrial sector based on the input-output model. According to China's 2017 National Industry Classification Standard [30], this study

separates industrial export products into different industrial sectors. When considering the environmental pollution from producing industrial products, we selected emissions of wastewater, waste gas, and solid waste as indicators of environmental pollution. The industrial sector's direct pollution coefficient is:

$$D_i = E_i / P_i \tag{6}$$

where $D_i$ is the direct emission coefficient of industry sector $i$, $E_i$ is the volume of pollution emissions in sector $i$, and $P_i$ is the total industrial output of sector $i$.

The industrial sector's indirect emission coefficient is:

$$C_i = \sum D_i b_{ij} \tag{7}$$

where $D_i$ is the direct consumption coefficient, and $b_{ij}$ is the complete consumption coefficient.

The industrial sector's complete pollution coefficient is:

$$T_i = D_i + C_i = D_i (I - A)^{-1} \tag{8}$$

## 3.3 Calculation of the pollution coefficient for export trade

The production of export products is provided, in part, domestically and with imported intermediate products, while the imported intermediate products are produced abroad. Therefore, when calculating a country's pollution emissions caused by the final use of export products, imported intermediate products should be excluded. We use $A_{im}$ to indicate the imported intermediate product and $A_d$ as the domestic input; then, $A = A_{im} + A_d$. When considering the industrial sector's environmental pollution impacts on the production of domestic products, the complete pollution coefficient should be:

$$T_i = D_i + C_i = D_i (I - A_d)^{-1} \tag{9}$$

This study follows works by Shen et al. [26] and Chen et al. [24] to exclude imported intermediate inputs. Assuming that matrix $M$ exists, or $A_{im} = M \times A$, then the domestic intermediate product input is $A_d = (I–M)A$, where $M$ denotes the matrix of import coefficients, assuming that each department inputs the same proportion of intermediate import products to all other departments, and $M$ is a diagonal matrix. Specifically, this is expressed as:

$$m_{ij} = IM_i / (X_i + IM_i - EX_i) \tag{10}$$

where $i,j = 1,2,3,\ldots,n$; when $i \neq j$, $m_{ij} = 0$. Further, $m_{ij}$ is a diagonal matrix element, and $IM_i$ and $EX_i$ represent the import and export values of department $i$, respectively.

$$M = \begin{pmatrix} m_{11} & \cdots & m_{1n} \\ \vdots & \ddots & \vdots \\ m_{m1} & \cdots & m_{mm} \end{pmatrix} \tag{11}$$

Therefore, the full emissions coefficient of the pollutants within a country's export trade is:

$$T_i = D_i + C_i = D_i (I - (I - M) A)^{-1} \tag{12}$$

The pollutant emissions within the export trade are:

$$E_i = T_i \cdot EX_i \tag{13}$$

### 3.4 Analysis of the environmental effects of Guangdong's industrial product export trade

We adopt Grossman et al.'s [2] classical perspective to analyze trade's impact on the environment given scale, structural, and technical effects using the LMDI method. First, we used $E_0$ to denote the industrial pollution emissions in the base period and $E_t$ as the industrial pollution emissions in the reporting period. The total effect of changes in pollution emissions during the period is composed of the scale effect $E_s$, structural effect $E_c$, and technical effect $E_T$:

$$\Delta E = E_t - E_0 = E_s + E_c + E_T \tag{14}$$

According to the decomposition of the LMDI index, the scale, structural, and technical effects of industry $i$'s embodied pollutant emissions are as follows:

$$E_{is} = U_i \cdot \ln \left(\frac{EX_{it}}{EX_{i0}}\right) \tag{15}$$

$$E_{ic} = U_i \cdot \ln \left(\frac{e_{it}}{e_{i0}}\right) \tag{16}$$

$$E_{iT} = U_i \cdot \ln \left(\frac{T_{it}}{T_{io}}\right) \tag{17}$$

$$U_i = \frac{E_{it} - E_{i0}}{lnE_{it} - lnE_{i0}} \tag{18}$$

where $EX_{it}$ denotes the export value in the reporting period, and $EX_{i0}$ is the export value in the base period. The change in the export value indicates the degree of change in the export scale; $e_{it}$ and $e_{i0}$ are the ratios of the export value of industry $i$ in the reporting and base periods, respectively; and the change in the ratio indicates the change in the export industry structure. Moreover, $T_{it}$ and $T_{i0}$ are the complete emission coefficients for the reporting and base periods, respectively. The change in the complete emission coefficient indicates a technology change.

### 3.5 Data selection

Data on Guangdong Province's foreign trade were derived from the *Guangdong Statistical Yearbook* [27]. The input-output table was obtained from the *National Bureau of Statistics* [28], and the various industries' pollution emission data were obtained from the *China Environmental Statistics Yearbook* [29]. This study classified Guangdong's foreign trade industries according to the 2017 National Industry Classification Standard [30]. The exported industrial products were classified into 19 industries, noted in the following Table 1.

## 4. Results and analysis

### 4.1 Differences in pollution coefficients of industrial sectors

Considering the emission of the whole production process, whether it is the emission of wastewater, waste gas, or solid waste, the complete emission coefficient is much larger than the direct emission coefficient. In terms of emission trends, both the complete emission coefficient and the direct emission coefficient show a downward trend, indicating that the pollution emission per unit of output is decreasing; however, the pollution emissions of different industries are different. Table 2 lists the top 5 industries with the highest emission coefficient.

**Table 1. Industries of Guangdong Province's export products.**

| Codename | Export product industry | Codename | Export product industry |
|---|---|---|---|
| C1 | Electrical machinery and equipment manufacturing | C11 | Transportation equipment manufacturing |
| C2 | Textile and apparel | C12 | General equipment manufacturing |
| C3 | Textiles | C13 | Culture and education, aerobics, sports, and entertainment goods manufacturing |
| C4 | Non-metallic mineral products | C14 | Rubber and plastic products |
| C5 | Ferrous metal smelting and rolling processing | C15 | Pharmaceutical manufacturing |
| C6 | Chemical raw materials and chemical products manufacturing | C16 | Instrumentation manufacturing |
| C7 | Computer, communication, and other electronic equipment manufacturing | C17 | Non-ferrous metal smelting and rolling processing |
| C8 | Furniture manufacturing | C18 | Paper and paper products |
| C9 | Metal products | C19 | Special equipment manufacturing |
| C10 | Leather, fur, feathers, and related products, and shoe manufacturing | | |

## 4.2 Complete emissions from the industrial sector

From the perspective of the total emissions trend, the total emissions of waste gas and solid waste are on the rise; the total emissions of wastewater has little change. It shows that the management of industrial wastewater emissions has been controlled well in recent years, but control of waste gas and solid waste emissions needs to be strengthened. However, whether it is the total emissions of waste gas, wastewater or solid waste, the top five industries with the highest emissions account for about 80% of the total discharge of all industrial industries. Therefore, the pollution emissions reflects a relatively severe industry agglomeration. Table 3 lists top 5 industries with complete emissions.

## 4.3 Decomposition of export emissions' influencing factors

We used the Logarithmic Mean Divisia Index (LMDI) to analyze waste gas emissions' influence on the scale, structural, and technical effects and observed that the total waste gas emissions effect is essentially positive. Therefore, except for a few industries, emissions have decreased. The industries with the highest total effect on waste gas emissions are C7, C4, and C1; these three industries primarily exhibit the most apparent increase in waste gas emissions due to the scale effect's increase. The technical effects' positive impact on waste gas emissions cannot sufficiently offset the increase in emissions created by the increase in scale. Any changes in structural effects in terms of waste gas emissions were unclear. Fig 2 shows the decomposition of factors affecting embodied waste gas emissions.

**Table 2. Top 5 industries with the highest emission coefficient.**

| Rank | Waste gas (unit: cubic meter/10,000 RMB) | | Wastewater (unit: ton/10,000 RMB) | | Solid waste (unit: ton/10,000 RMB) | |
|---|---|---|---|---|---|---|
| | Direct emission coefficient | Complete emission coefficient | Direct emission coefficient | Complete emission coefficient | Direct emission coefficient | Complete emission coefficient |
| 1 | C5 (28376.368) | C17 (31010.096) | C18 (16.649) | C18 (25.903) | C5 (0.697) | C6 (7.166) |
| 2 | C4 (20785.254) | C4 (29520.253) | C3 (4.678) | C6 (15.541) | C6 (0.394) | C5 (2.841) |
| 3 | C17 (8564.146) | C5 (10409.463) | C6 (3.079) | C15 (10.441) | C17 (0.284) | C18 (1.206) |
| 4 | C18 (4682.900) | C1 (7820.559) | C5 (2.598) | C3 (10.296) | C18 (0.158) | C17 (1.154) |
| 5 | C6 (4414.316) | C3 (7513.756) | C15 (2.069) | C5 (9.315) | C4 (0.126) | C12 (0.794) |

**Table 3. Top 5 industries with complete emissions.**

| Rank | Complete emissions of Waste gas (unit: cubic meter) | Complete emissions of wastewater (unit: ton) | Complete emissions of solid waste (unit: ton) |
|---|---|---|---|
| 1 | C7 (4507.227) | C7 (20412.443) | C7 (2963.651) |
| 2 | C1 (2580.513) | C3 (10727.295) | C1 (1888.020) |
| 3 | C4 (1278.829) | C1 (6394.221) | C5 (881.857) |
| 4 | C8 (1016.889) | C10 (6073.984) | C10 (477.338) |
| 5 | C3 (782.823) | C2 (3662.598) | C8 (383.715) |
| Total emissions of the top five industries | 10166.282 | 47270.54 | 6594.581 |
| Total emissions of all industrial sectors | 12595.249 | 59585.401 | 8395.637 |
| Percentage of top 5 total emissions | 80.72% | 79.33% | 78.55% |

Decomposing the influencing factors of wastewater emissions reveals that the scale effect of wastewater emissions is mostly negative. The three largest industries with positive scale effects are C7, C5, and C18. Regarding C7, the technical and structural effects are all negative, but they cannot offset the increased effect caused by the scale effect. Although the technical effects for C5 and C18 decrease wastewater emissions, both the scale and structural effects increase wastewater emissions; these technical effects cannot offset the scale and structural effects. While most industries' structural and technological effects exhibit reduced emissions, they cannot offset the scale effects. Fig 3 shows the decomposition of factors affecting wastewater emissions.

Decomposing the solid waste emissions indicated that the three largest industries with positive scale effects are C1, C5, and C7. The scale and structural effects of C1 and C5 are both positive, and the technical effect is negative. The positive impact of technological improvement on the environment cannot offset the negative environmental impact due to the expansion of scale and the increase in the proportion of the industrial sector. Regarding C7, the scale effect was positive, and the structural and technical effects were negative. The decline in the proportion of structural effects and improvement in technology cannot offset the increase in solid waste emissions caused by the increase in scale. The total effect of solid emissions has increased among most industries; in most cases, their structural or technical effects are insufficient to

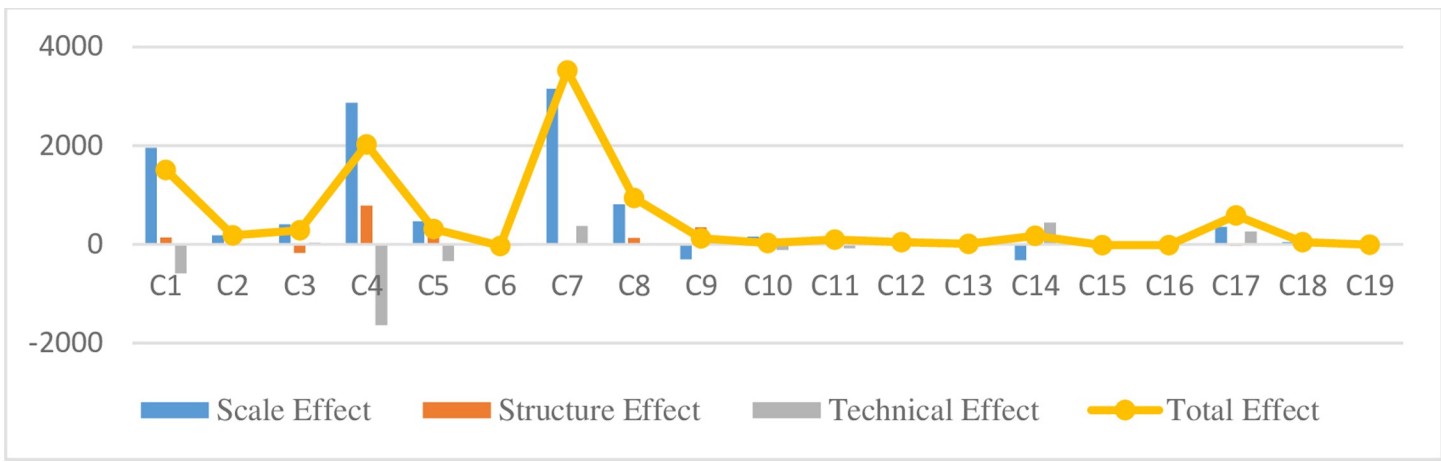

**Fig 2. Decomposition of factors affecting embodied waste gas emissions (unit: 100 million cubic meters).**

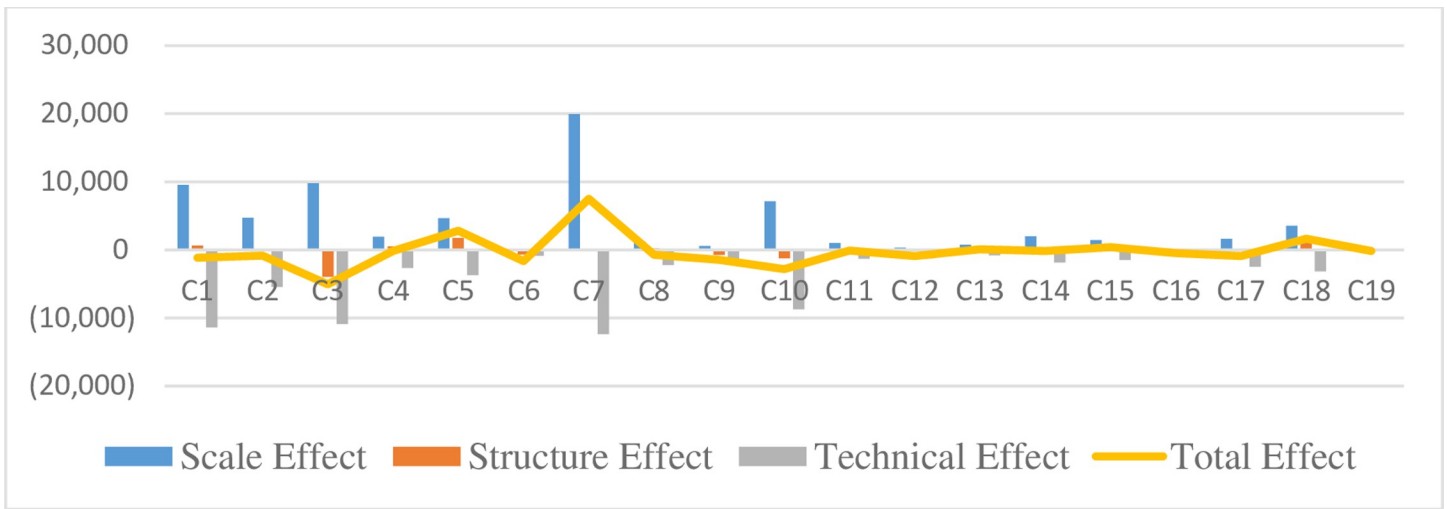

**Fig 3. Decomposition of factors affecting wastewater emissions (unit: 10,000 tons).**

offset the negative scale effects. Fig 4 shows the decomposition of the influencing factors of solid waste emissions.

We considered this data in analyzing several industries with more severe pollution. We discovered that the total effect of the emissions from waste gas and solid waste are positive for C1 and C4; while all wastes' emissions were positive for C5, C7, and C18, the wastewater and solid waste emissions had greater total effects for C5. Regarding C7, all three pollutants had a greater total effect, while wastewater emissions had a greater total effect on C18.

Regarding C1 and C4, the total emissions from waste gas and solid waste are positive, but their total effect on wastewater is negative, indicating that these industries should focus on strengthening the treatment of waste gas and solid waste. In terms of the emissions from waste gas and solid waste, their technical effects cannot offset their scale and structural effects. However, wastewater emissions' technical effect can offset the increase in pollution caused by the scale and structural effects.

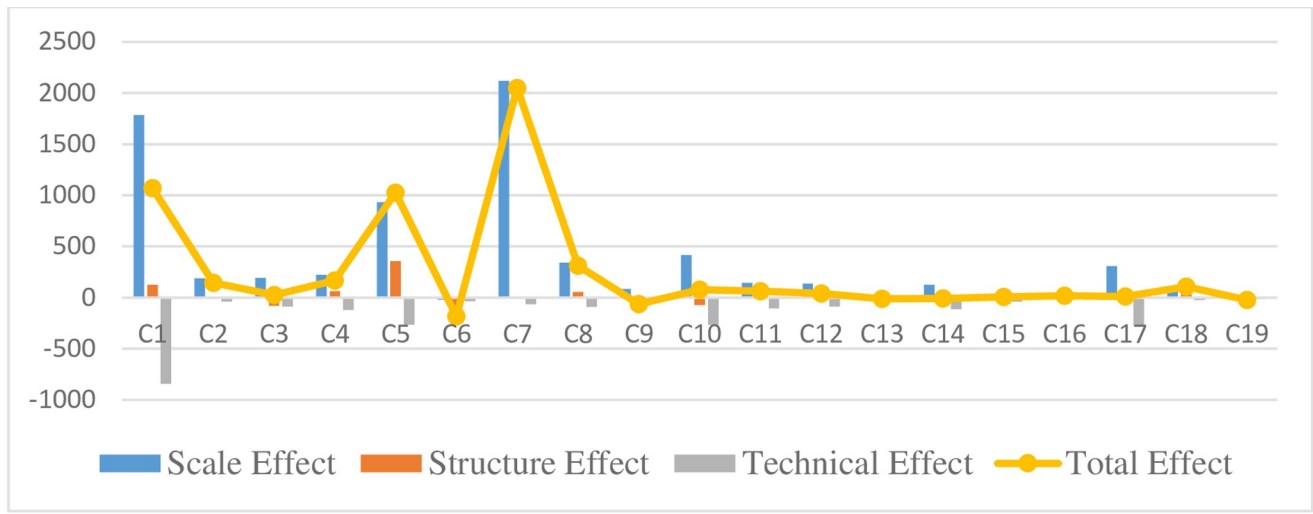

**Fig 4. Decomposition of the influencing factors of solid waste emissions (unit: 10,000 tons).**

The emissions observed in C5, C7, and C18 wastes increased, with such increases for C5 and C18 primarily due to the simultaneous increase in the scale and structural effects; the technological effect cannot offset the impacts of the scale and structural effects. The structural and technical effects in industry C7 have improved, but cannot offset the scale effect's impacts.

## 5. Conclusion

The pollution emitted from exporting industrial products in Guangdong Province is generally increasing. According to industry data, scale has a greater impact on pollution emissions, and clear effects can be observed from decreasing pollution. The overall trend of wastewater emissions remains stable because technological effects bring environmental improvements, while structural effects play a certain role. Obviously, the technical level improvement can bring about the improvement of pollution emission significantly.

Important industries that produce pollution emissions include the computer, communications, and other electronic equipment-manufacturing industries; electrical machinery and equipment manufacturing industry; ferrous metal smelting and rolling processing industry; and the non-ferrous metal smelting and rolling processing industry. Emissions of the top five industries account for about 80% of total emissions. The government of Guangdong Province should strengthen its environmental controls over the production processes for these key industry products and related inputs and actively promote the reduction of pollution in these industries.

In the context of developing a green economy, the government actively promotes improvement of technical level and conducts a targeted treatment of industries with high pollution emissions. As technical effects have a greater impact on pollution emissions, important ways to reduce pollution involve increasing R&D efforts to increase the added value of products and develop high-quality foreign trade. Fundamental ways to address environmental issues could also include strengthening improvements in the emissions trading market, further incentives for companies to pay for environmental pollution, and internalizing environmental costs. To better participate in international trade, it is important to actively encourage relevant enterprises to participate in environmental management system certification. On the one hand, EMS is conducive to domestic pollution reduction, and on the other hand, it can also enhance the environmental competitiveness of products worldwide.

This article only examined emissions data for wastewater, waste gas, and solid waste by industry up to 2015 due to changes in the caliber of the *Statistical Yearbook*. The selected data are total emissions data, as the specific pollution emissions and their impacts on the environment differ across industries. In the future, we will continue to explore the environmental impact of specific types of foreign trade-based pollution. Further, this study's research remains at the industry level and does not consider the heterogeneity of corporate emissions; this is an important direction to pursue for future studies.

## Supporting information

**S1 File.**
(DOCX)

**S1 Data.**
(ZIP)

## Acknowledgments

We would like to thank Editage (www.editage.com) for English language editing.

## Author Contributions

**Conceptualization:** Yu-Xia Du, Ming-Jie Li.

**Data curation:** Yu-Xia Du.

**Formal analysis:** Yu-Xia Du.

**Investigation:** Ming-Jie Li, Jun-Jie Huang.

**Methodology:** Yu-Xia Du, Ming-Jie Li.

**Project administration:** Yu-Xia Du, Ming-Jie Li.

**Writing – review & editing:** Jun-Jie Huang.

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
