## [Decision Letter · Decision Letter 0]

7 Jun 2022

PONE-D-22-05437Research on the emissions from industrial products exported from Guangdong Province—an input-output model analysisPLOS ONE

Dear Dr. Ming Jie Li,

Thank you for submitting your manuscript to PLOS ONE. After careful consideration, we feel that it has merit but does not fully meet PLOS ONE’s publication criteria as it currently stands. Therefore, we invite you to submit a revised version of the manuscript that addresses the points raised during the review process.

We look forward to receiving your revised manuscript.

Kind regards,

Balasubramani Ravindran, Ph.D

Academic Editor

PLOS ONE

Journal Requirements:

4. Please ensure that you refer to Figure 2-13 in your text as, if accepted, production will need this reference to link the reader to the figure.

Reviewers' comments:

Reviewer's Responses to Questions

**Comments to the Author**

1. Is the manuscript technically sound, and do the data support the conclusions?

Reviewer #1: Yes

Reviewer #2: Yes

Reviewer #3: No

2. Has the statistical analysis been performed appropriately and rigorously? 

Reviewer #1: Yes

Reviewer #2: Yes

Reviewer #3: Yes

3. Have the authors made all data underlying the findings in their manuscript fully available?

Reviewer #1: Yes

Reviewer #2: No

Reviewer #3: Yes

4. Is the manuscript presented in an intelligible fashion and written in standard English?

Reviewer #1: Yes

Reviewer #2: Yes

Reviewer #3: Yes

5. Review Comments to the Author

Reviewer #1: The author has highlighted only about waste water emissions what about other pollutants from the industries.

I doubt that adjustment in industrial structure would reduce the emission.

Why the author has not insisted about EMS certification as a control measure

Reviewer #2: This study investigated the wastewater, waste gas, and solid waste emissions in Guangdong’s industrial exports from 2004 to 2015. The authors use LMDI index and input-output model to analyze the factors. This study classified Guangdong’s foreign trade industries and analyze and compare waste gas emissions, wastewater emissions, solid waste emissions for each category. Overall, the authors have spent a lot of efforts on this study and the manuscript has been prepared in a professional manner. It is well organized and the message is clear. However, there are areas for improvement in terms of content. The current manuscript has a major revision of some issues that need to be addressed before being considered for publication.

1. The unit in figure 1 is not defined,

The manufactured good amount and total export is seemed equal in figure however the proportion of manufactured goods have fluctuation, it’s better that chat shows the differences.

2. The second section (research area and method) lacks a detailed review of the model and methodology

What is the other method and index to analyze the problem?

What is the other methods and index that used in existing literature?

Why authors use input-output model and LMDI to analyze the problem?

3. Most of research in literature review is focused on research conducted in china, it’s proposed to mention and investigate other existing research in other countries.

4. Some parts of the paper can be shorten (e.g., Sections 4) and proposed to summarize some analysis and result in table for readers.

Reviewer #3: 1. Keywords should ideally be phrases of 2-4 words. Research keywords are not carefully selected. It is also necessary to give the reader an adequate explanation after choosing a keyword. and change the keywords "LMDI" to "The logarithmic mean Division index".

2. The reason for doing research (research gap) and research innovation must be clearly stated in the abstract and the introduction.

3. It is better to compare the results of this study with other similar studies. Also, the benefits of research are described.

6. PLOS authors have the option to publish the peer review history of their article (what does this mean?). If published, this will include your full peer review and any attached files.

Reviewer #1: No

Reviewer #2: No

Reviewer #3: **Yes: **Ali Reza Afshari

---

## [Author Response · Author response to Decision Letter 0]

13 Sep 2022

Reviewer #1: The author has highlighted only about waste water emissions what about other pollutants from the industries.

I doubt that adjustment in industrial structure would reduce the emission.

Why the author has not insisted about EMS certification as a control measure

Response:

This study uses an input-output model to analyze the wastewater, waste gas, and solid waste emissions found in Guangdong’s industrial exports from 2004 to 2015.We studied not only waste water emissions, but also waste gas, and solid waste emissions.

1.This paper carries out the structural decomposition of the emissions from industrial products exported from Guangdong Province. We suggest that improvement of technical level will reduce the emission significant.

2.On the page14, we add EMS as a control measure to reduce emissions for international trade.

Reviewer #2: This study investigated the wastewater, waste gas, and solid waste emissions in Guangdong’s industrial exports from 2004 to 2015. The authors use LMDI index and input-output model to analyze the factors. This study classified Guangdong’s foreign trade industries and analyze and compare waste gas emissions, wastewater emissions, solid waste emissions for each category. Overall, the authors have spent a lot of efforts on this study and the manuscript has been prepared in a professional manner. It is well organized and the message is clear. However, there are areas for improvement in terms of content. The current manuscript has a major revision of some issues that need to be addressed before being considered for publication.

1. The unit in figure 1 is not defined,

The manufactured good amount and total export is seemed equal in figure however the proportion of manufactured goods have fluctuation, it’s better that chat shows the differences.

2. The second section (research area and method) lacks a detailed review of the model and methodology

What is the other method and index to analyze the problem?

What is the other methods and index that used in existing literature?

Why authors use input-output model and LMDI to analyze the problem?

3. Most of research in literature review is focused on research conducted in china, it’s proposed to mention and investigate other existing research in other countries.

4. Some parts of the paper can be shorten (e.g., Sections 4) and proposed to summarize some analysis and result in table for readers.

Response:

 1.We add the unit for figure 1. As the proportion of exported industrial products is greater than 97%. Actually, the amount of manufactured goods is really close to the amount of total export. we just can adjust the Figure 1 to make it more clear.

2.We add detailed review of the model and methodology in the second section which can be seen in P4-P6. 

Most papers use input-output approach to analyze this problem. SRIO(Single regional Input-output)\\BTIO(Bilateral trade input-output)\\MRIO(Multilateral regional input-output) are widely used, this paper just studied the problem of Guangdong Province, so we just used SRIO to analysis the emission of industrial products. SDA(Structural Decomposition Analysis) and LMDI(The logarithmic mean Division index) are widely used to describe the decomposition of pollution emission structure, but LMDI can overcome the cross-term problem of SDA well, so we chose LMDI as the index. Based on your comments, we have clarified and explained the difference of the current study with the available literature, which can be seen in P4-P6

3.We also list some existing research in other countries like OECD countries\\American\\ Norway\\ South Korea\\ Poland\\ Turkish\\ Saudi Arabia, and so on, which can be seen in P4-P5

4. Based on your comments, Sections 4 has been shorten and we summarized analysis and result in table, which can be seen in P11 and table2-3.

Reviewer #3: 1. Keywords should ideally be phrases of 2-4 words. Research keywords are not carefully selected. It is also necessary to give the reader an adequate explanation after choosing a keyword. and change the keywords "LMDI" to "The logarithmic mean Division index".

2. The reason for doing research (research gap) and research innovation must be clearly stated in the abstract and the introduction.

3. It is better to compare the results of this study with other similar studies. Also, the benefits of research are described.

Response:

1. Based on your comments, we have selected research keywords which can be seen P2.

 2. We have clarified and explained reason for doing research which can be seen in P3. Research innovation has been stated in the introduction which can be seen P3-P4.

 3. Due to the different regions studied, we made some comparisons in terms of research methods, which can be seen in p6 Based on your comments, the benefits of research are described in P4.

---

## [Editor Report · Decision Letter 1]

19 Sep 2022

PONE-D-22-05437R1Research on the emissions from industrial products exported from Guangdong Province--an input-output model analysisPLOS ONE

Dear Dr. Li,

Thank you for submitting your manuscript to PLOS ONE. After careful consideration, we feel that it has merit but does not fully meet PLOS ONE’s publication criteria as it currently stands. Therefore, we invite you to submit a revised version of the manuscript that addresses the points raised during the review process.

We look forward to receiving your revised manuscript.

Kind regards,

Muhammad Ikram

Academic Editor

PLOS ONE

Journal Requirements:

Additional Editor Comments:

Dear Authors,

The paper is improved but sincerely the conclusions are not clear. I suggest to enhance the entire-section providing managerial implications, main issues and what are the key-messages of your work.

---

## [Author Response · Author response to Decision Letter 1]

28 Sep 2022

Journal Requirements:

Response:

We reviewed our reference carefully, and we haven’t find any papers have been retracted.

Dear Authors,

The paper is improved but sincerely the conclusions are not clear. I suggest to enhance the entire-section providing managerial implications, main issues and what are the key-messages of your work.

Response:

We read our paper carefully and improved it as much as we could. As for the deeper causes of the described problems, we may need to conduct further research and publish in the next paper.

Response:

Thank you for your valuable advice. We used PACE to correct our pictures, which really helped us a lot.

---

## [Editor Report · Decision Letter 2]

5 Oct 2022

Research on the emissions from industrial products exported from Guangdong Province--an input-output model analysis

PONE-D-22-05437R2

Dear Dr. Li,

We’re pleased to inform you that your manuscript has been judged scientifically suitable for publication and will be formally accepted for publication once it meets all outstanding technical requirements.

Kind regards,

Muhammad Ikram

Academic Editor

PLOS ONE
---

## [Editor Report · Acceptance letter]

26 Oct 2022

PONE-D-22-05437R2 

Research on the emissions from industrial products exported from Guangdong Province—an input-output model analysis 

Dear Dr. Li:

I'm pleased to inform you that your manuscript has been deemed suitable for publication in PLOS ONE. Congratulations! Your manuscript is now with our production department. 

Kind regards, 

on behalf of

Professor Muhammad Ikram 

Academic Editor

PLOS ONE